# A reporter system coupled with high-throughput sequencing unveils key bacterial transcription and translation determinants

Eva Yus [1,2], Jae-Seong Yang[1,2], Adrià Sogues[1,2,3] & Luis Serrano[1,2,4]

Quantitative analysis of the sequence determinants of transcription and translation regulation is relevant for systems and synthetic biology. To identify these determinants, researchers have developed different methods of screening random libraries using fluorescent reporters or antibiotic resistance genes. Here, we have implemented a generic approach called ELM-seq (expression level monitoring by DNA methylation) that overcomes the technical limitations of such classic reporters. ELM-seq uses DamID (*Escherichia coli* DNA adenine methylase as a reporter coupled with methylation-sensitive restriction enzyme digestion and high-throughput sequencing) to enable in vivo quantitative analyses of upstream regulatory sequences. Using the genome-reduced bacterium *Mycoplasma pneumoniae*, we show that ELM-seq has a large dynamic range and causes minimal toxicity. We use ELM-seq to determine key sequences (known and putatively novel) of promoter and untranslated regions that influence transcription and translation efficiency. Applying ELM-seq to other organisms will help us to further understand gene expression and guide synthetic biology.

[1] Centre for Genomic Regulation (CRG), The Barcelona Institute for Science and Technology, Doctor Aiguader 88, Barcelona 08003, Spain. [2] Universitat Pompeu Fabra (UPF), Barcelona Spain. [3] Institut Pasteur, Unité de Microbiologie Structurale (CNRS) UMR 3528, Université Paris Diderot, 25 rue du Docteur Roux, Paris 75724, France. [4] Institució Catalana de Recerca i Estudis Avançats (ICREA), Pg. Lluis Companys 23, Barcelona 08010, Spain. Eva Yus and Jae-Seong Yang contributed equally to this work.  Correspondence and requests for materials should be addressed to L.S. (email: luis.serrano@crg.eu)

Understanding the sequence-dependent mechanisms that regulate gene expression is fundamental for generating accurate predictive cell models[1], and for developing biotechnological applications. Unfortunately, however, as there are a number of factors that can influence protein levels, it is highly challenging to determine the contribution of such upstream sequences. This is especially true when only having information about an organism's genome, or even when knowing endogenous protein levels from *Omics* data.

In particular, transcript levels in bacteria are mainly determined by the rate of transcription, especially by that of the initiation step[2, 3]. The promoter sequences determine the rate of transcription initiation by modulating the binding affinity of RNA polymerase[4], the transition from a closed to open complex[5], as well as the stability of the open complex itself[6]. Although the general features of the house-keeping sigma 70-dependent promoters are similar across bacteria, it was recently shown that sequence determinants have different weighs even in related bacteria[7, 8]. In addition, the GC content can also affect the nucleotide propensity of promoters[9, 10]. These differences can explain how the same synthetic promoters result in weakly correlated gene expression patterns between *Synechococcus* sp. Strain PCC 7002 and *Escherichia coli*[11]. In this way, promoter predictors trained with certain bacteria could result in inaccurate predictions when applied to other bacteria[12, 13], thereby limiting their quantitative prediction capability.

Translation control on the other hand, despite being equally crucial for tailored gene expression, is less understood and predictable than transcription. Although many mRNAs in bacteria contain in their 5′-untranslated region (UTR) a Shine-Dalgarno (SD) motif complementary to the 16S ribosomal RNA, there are numerous examples of mRNAs that start at the ATG[14],

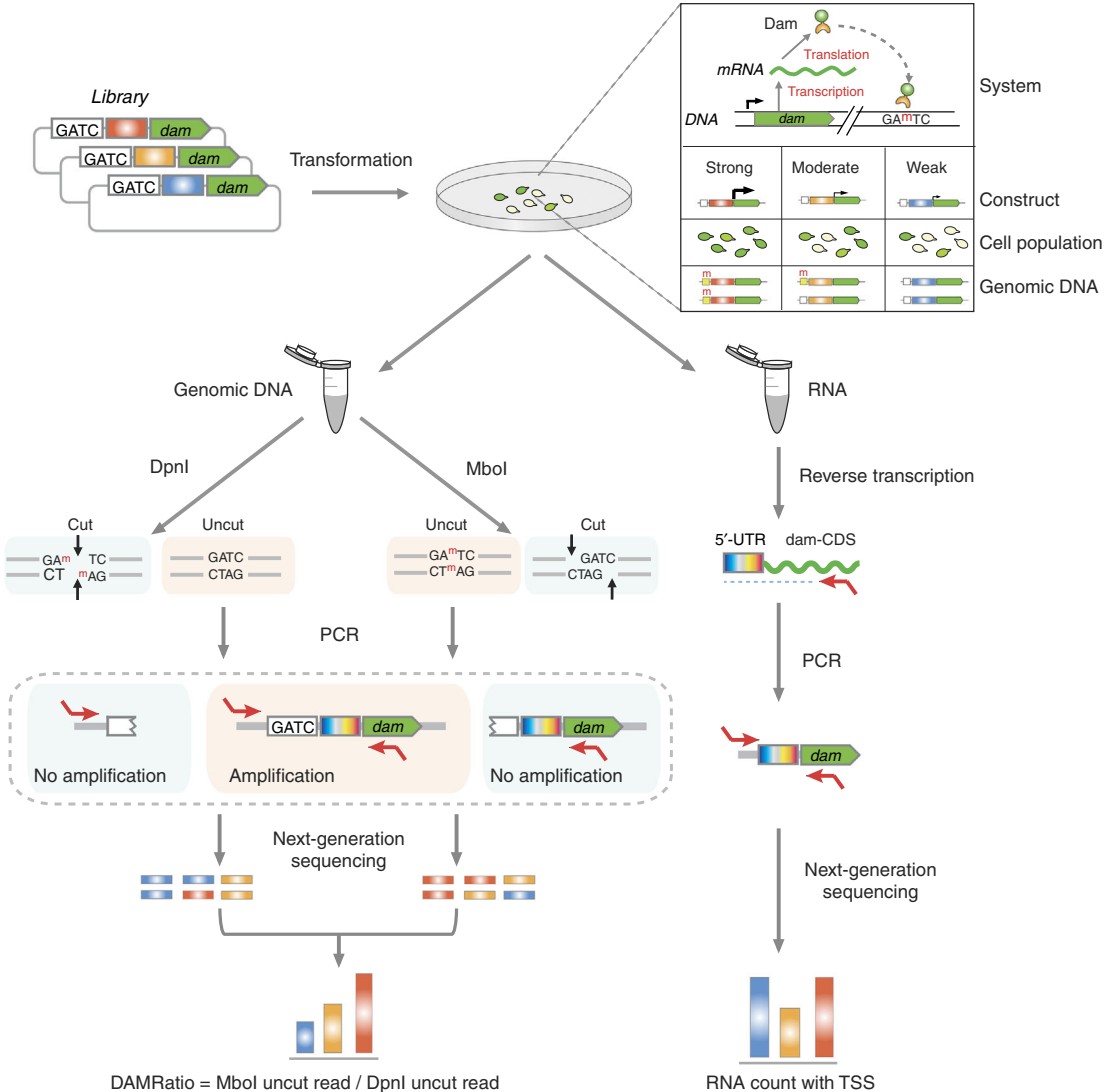

**Fig. 1** Workflow of ELM-seq—a DNA methylation-based mutational screen of promoters and 5′-UTRs. The expressed Dam protein transfers a methyl group onto the adenine in the GATC motif. Transcription and translation efficiencies control the expression of the Dam protein and consequently the level of adenine methylation. We developed this protocol in *M. pneumoniae* as follows: we transformed a library of promoter/5′-UTR random constructs containing four GATC sites into *M. pneumoniae*. We collected genomic DNA and RNA separately. The genomic DNA was digested separately using two distinct methylation-sensitive enzymes (MboI that digests unmethylated GATC sites and DpnI that digests methylated GATC sites). Then we performed PCR reactions that only amplify uncut DNA library fragments, and, in each case, counted the number of reads obtained from next-generation sequencing to determine the so-called DAMRatio. The DAMRatio represents the ratio of the uncut read frequency in the MboI digestion pool to that in the DpnI digestion pool for each random sequence. In the case of mRNAs, the number of reads and TSS were determined for each transcript of our libraries by Dam-specific PCR reactions

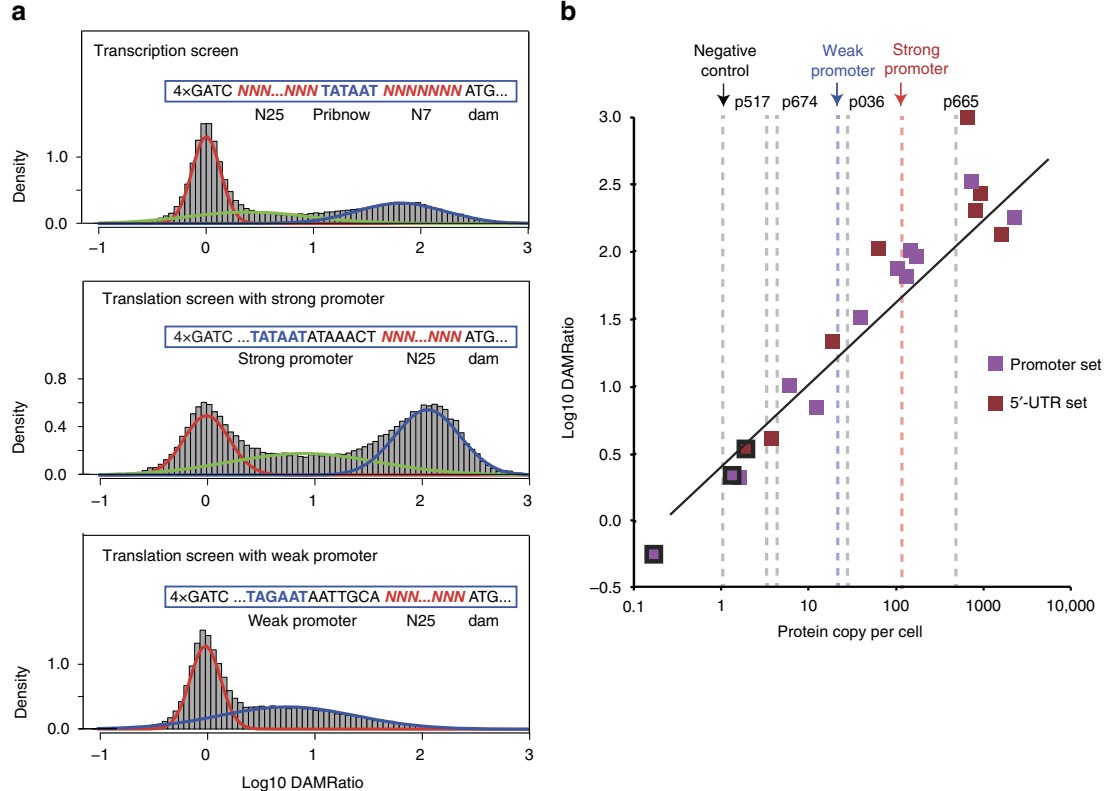

**Fig. 2** Cassette design for screening three different libraries and the relationship between Dam copy number and DAMRatio. **a** Design of the promoter and 5′-UTR constructs and their observed DAMRatio distributions. The transcription screen had a total of 82,639 promoter sequences, while the translation screen had 94,945 and 71,478 5′-UTR sequences for the strong and weak promoters, respectively. The sequence details of the constructs are shown inside the *blue box*. Randomized sequence positions are indicated by Ns and the numbers below the *blue boxes* indicate how many random positions are included. In all cases, four GATC sites are located upstream of the promoter regions. To compare the log-transformed DAMRatios among different experiments, we standardized them by setting the left-most peak to zero. The fitted Gaussian mixture distributions are indicated as *red, green*, and *blue lines* (see Methods). The *blue* and *red lines* indicate the probability of being highly and lowly productive sequences, respectively. **b** Relationship between DAMRatio and Dam protein copy number. Twelve different promoter sequences and eight 5′-UTR sequences were selected from the different random libraries based on their DAMRatios to cover a broad range of Dam activities. The protein copy numbers of four natural promoters (p517, p674, p036, and p665; see Methods) and strong and weak promoters are depicted as *dashed lines*. Dam protein copy numbers were estimated with regression models generated by published proteome-wide copy number data[30] and mass spectrometry data. The *squares* outlined in *black* represent clones that were not detected by mass spectrometry. Their protein copy numbers were estimated from the DAMRatio using linear interpolation and were not used in the correlation analysis. The overall Pearson's correlation coefficient (*r*) between protein copy number and the DAMRatio is 0.96

or do not have a SD sequence at all[15]. In addition, Gram-positive bacteria have a variable optimum space distance between the SD and the translation start codon (TSC)[16]. Recently, it was shown that the 5′-UTR secondary structure also has an impact on translation efficiency. Current algorithms that take this into consideration[17–19] are optimized for *E. coli*.

Screening methodologies that combine the expression of fluorescent proteins (FP) or antibiotic resistance markers under the control of randomized sequences have previously been developed[20–22]. However, most of these methods have some limitations. Growth-based selection using antibiotics for instance has a number of drawbacks including a relatively low sensitivity/dynamic range[22]. Some cells can develop survival mechanisms, leading to false-positives. Moreover, this approach cannot identify negative clones that are not viable[22]. The common alternative, FP selection by cell sorting, is laborious and less reproducible[23]. In fact, as the intensity of the commonly used green fluorescent protein (GFP) is at its best in high oxygen levels and at a neutral pH, often only moderate fluorescence is obtained in bacteria growing under anaerobic conditions[24]. In addition, there are limitations with regard to cell aggregation and certain cell shapes and sizes[25].

Recently, a technique called DamID, which is based on *E. coli* DNA adenine methylase (Dam), has been established as an alternative method to chromatin immunoprecipitation (ChIP)[26]. When fused with a protein of interest (POI), such as a transcription factor, Dam methylates the GATC sequences of DNA that are in close proximity to the in vivo location of the POI. These DNA "footprints" are usually "measured" by differential digestion using methylation-sensitive restriction enzymes that cut the GATC sequence, and a high-throughput detection method, such as microarrays or DNA-seq[27]. Previous studies have shown that DamID qualitatively differentiates TF targets from nonspecific sites[28]. We thus hypothesized that the Dam enzyme could be used as a reporter system to monitor protein expression levels when combined with ultrasequencing, and termed this new, to the best of our knowledge, expression level monitoring technique ELM-seq.

We chose the genome-reduced bacterium *Mycoplasma pneumoniae* to test this system because it has a basic transcription (one house-keeping factor and one alternative sigma factor that is not expressed under normal conditions) and translation machinery[29]. However, despite extensive experimental analyses of these processes in this bacterium[30–32], we still do not fully

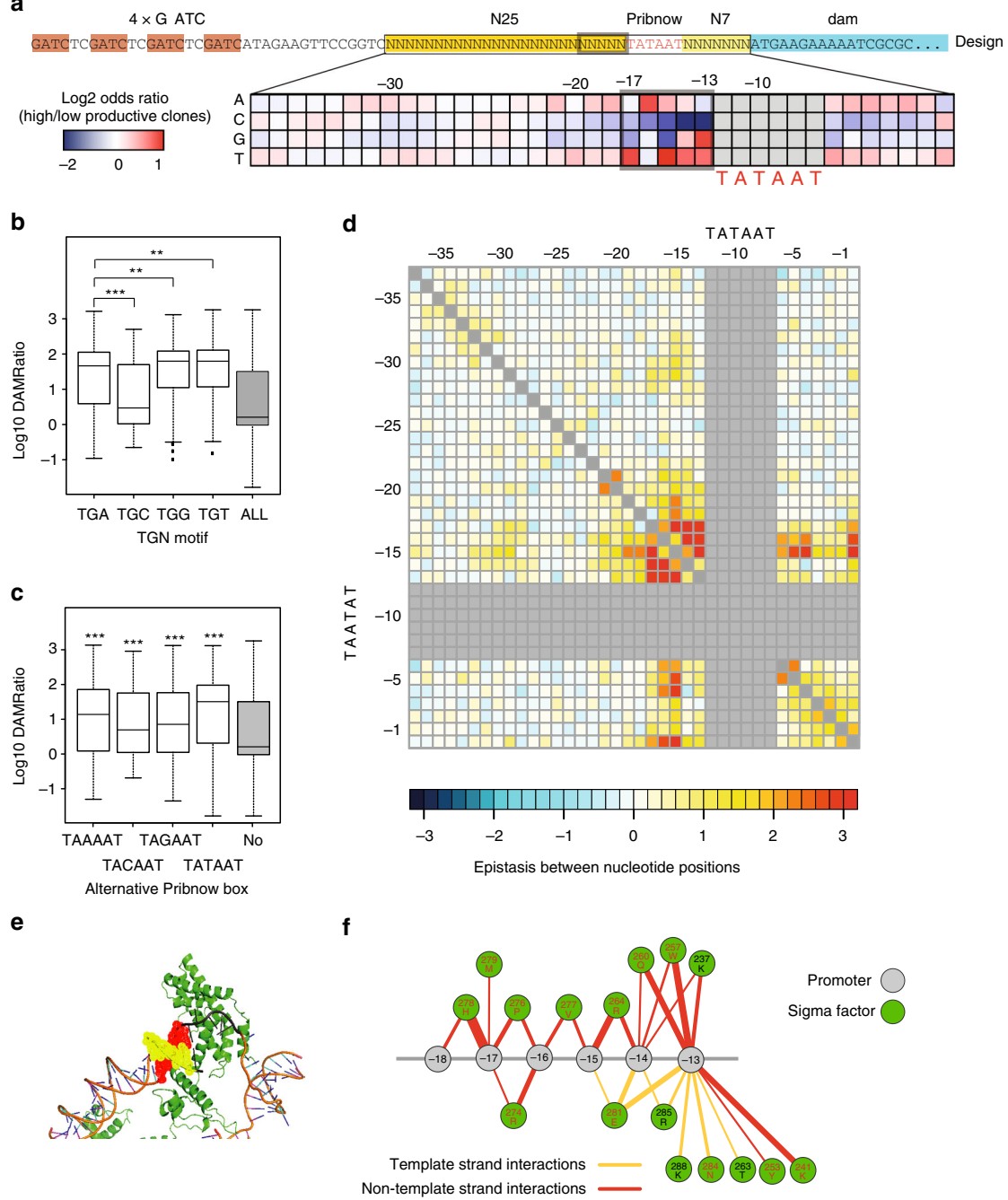

**Fig. 3** Deep sequence analysis of the promoter region of *M. pneumoniae*. **a** N25 and N7 depict the randomized regions around the Pribnow sequence (TATAAT). Log2 odds ratio shows the nucleotide bases between high-productive and low-productive promoters. **b** DAMRatios for the TG motif (positions −15 to −14 upstream of the Pribnow box) with different bases at position −13. ***$P$ value less than $2 \times 10^{-16}$ and **$P$ value is less than $1 \times 10^{-4}$. **c** The DAMRatios for additional alternative Pribnow motifs (TAAAAT, TACAAT, TAGAAT, and TATAAT) in the randomized promoters are significantly higher than the DAMRatio for having only one fixed TATAAT Pribnow box. ***$P$ value is less than $2 \times 10^{-16}$. **d** *Matrix* showing epistatic interactions between different bases of the randomized promoters. The promoter positions are from −37 to +1. Positive values mean significant epistatic interactions compared to the null model. **e** Sigma factor-promoter complex structure of *Thermophilus aquaticus* (PDB: 4XLN). Domain 2 and domain 3 of sigma factor interacting with the DNA sequence upstream of the Pribnow box. The sequence similarity between the solved structure and sigA (MPN352) of *M. pneumoniae* is over 90%. *Green color* depicts sigma factor, and *red* and *yellow dots* represent the −13 to −17 regions of the non-template and template strands of the promoter, respectively. **f** *Red* and *yellow edges* represent the interactions between sigma factor and the non-template and template strands of the promoter, respectively. *Edge widths* correspond to the number of atomic contacts (>5 Å). The −13 to −15 regions of both the template and non-template strands of the promoter interact with sigma factor. *Red font* represents the residues that are identical to the aligned sequence of the sigma factor of *M. pneumoniae*. $P$ values were calculated with a two-tailed $t$-test

understand how its gene expression is controlled. This hampers its use as a model organism in synthetic biology.

First, we have shown that Dam methylase activity is proportional to Dam protein levels, and that it has a good dynamic range (~10,000-fold). We have validated our approach confirming previously described *M. pneumoniae* promoter features[32], the preference for purine in the first base of the transcript[33], and that the bases around the +1 (transcription start site, TSS) of the transcript are important for determining the transcriptional outcome. We show that RNA secondary structure in the 5′-UTR is the main determinant for translation efficiency, with the SD sequence contributing to a lesser extent. We also found that frameshifts resulting from alternative TSCs in the 5′-UTR downregulate translation when the alternative ATG is nearby (up to 12 nucleotides (nts)) the original TSC. Using the sequence data, we have generated a model capable of predicting transcription and translation efficiencies. This model yielded fairly precise predictions for our library sequences (0.90 and 0.80 area under curve (AUC) for transcription and translation, respectively), and thus could be used in the future to guide new designs that incorporate repressors and activators. In conclusion, we have developed a straightforward gene expression reporter system that can easily be applied to other bacteria and eukaryotes in order to find and optimize sequence-based features affecting transcription and translation efficiencies.

## Results

**Dam as a reporter for *cis*-regulatory sequence screening**. A good reporter system should exhibit a wide dynamic range and employ a reporter whose activity is related to its abundance[34]. To show that Dam is a quantifiable reporter, we measured its activity in *M. pneumoniae* under the control of four known endogenous promoters (Supplementary Fig. 1, Supplementary Table 1, and Supplementary Data 1). As expected, we found a gradual variation of Dam activity measured by quantitative PCR (qPCR) and LC-MSMS (liquid chromatography–mass spectrometry; Supplementary Tables 2 and 3).

To screen for sequence determinants affecting transcription and translation efficiency, we constructed three different Dam reporter cassettes to drive *dam* expression (Supplementary Note 1). One consisted of a randomized promoter (transcription screen), while the other two had fixed promoters differing fivefold in expression level (determined by western blot and qPCR) and randomized 5′-UTR sequences (translation screens). All cassettes had four consecutive Dam methylation sites located on the 5′-end (4× GATC; Figs. 1 and 2a).

For each randomized construct, we estimated the expression level by calculating the DAMRatio (see Methods). The DAMRatio is determined by dividing the number of sequencing reads obtained after digesting genomic DNA with MboI (cuts GATC), by DpnI (G^mATC; Fig. 1). We performed two biological replicates, and as their DAMRatios were shown to be correlated ($r > 0.90$; $P < 10^{-10}$; two-tailed $P$ value for Pearson's correlation coefficient, Supplementary Fig. 2a–d and Supplementary Table 4) we combined them for analysis.

We obtained the DAMRatios for 82,639 promoter sequences, 94,945 and 71,478 5′-UTR sequences with the strong and weak promoters, respectively. The DAMRatios spanned a dynamic range of over 10,000-fold and showed a two-peak distribution (Fig. 2a and Supplementary Note 2). We will refer to sequences in the right peak as high-productive and to the left peak as low-productive. It is interesting that about two-thirds of the random promoter sequences belong to low-productive promoters even though they have an ideal Pribnow Box (TATAAT). This indicates that optimal sequence context around the Pribnow is

required for high levels of transcription[32]. As expected, in the case of the 5′-UTR library with the strong promoter we found a larger high-productive peak, while with the weak promoter it was more indistinct (Fig. 2a).

**Validation of the ELM-seq screens**. To validate the performance of our screening protocol, we randomly chose 12 promoter clones and eight 5′-UTR clones (with strong promoter) that spanned a wide range of DAMRatios (Supplementary Table 2), cloned and inserted them individually into *M. pneumoniae*, and measured Dam activity by qPCR (Supplementary Table 5). We also determined Dam protein copy number from a regression model using data derived from LC-MSMS[30] (Supplementary Data 2 and Supplementary Tables 5 and 6). We found that DAMRatios clearly correlate with protein copy numbers ($r = 0.96$; Fig. 2b; twofold error range[35]) and Dam activities ($r = 0.80$ and 0.90 for promoter and 5′-UTR screenings, respectively; Supplementary Fig. 2e, f) for all three screens. We also confirmed that the dynamic range of the DAMRatio for the selected cases agreed with that of qPCR experiments in both promoter and 5′-UTR studies ($r = 0.77$, $P = 2.5 \times 10^{-6}$; two-tailed $P$ value for Pearson's correlation coefficient, Supplementary Table 5). Thus, the probability of having a methylated GATC site is directly proportional to the average amount of Dam protein in the population. In the event of hemimethylation, GATCs cannot be cut by either DpnI or MboI. We simulated the effect of hemimethylation (Supplementary Note 2) and showed that it does not significantly affect the results.

Finally, we found that strong Dam expression did not significantly alter the proteome (Supplementary Fig. 3) or compromise the growth (Supplementary Table 7). Therefore, we are able to conclude that Dam is a quantifiable reporter capable of monitoring gene expression in vivo.

**Analysis of the promoter library reveals functional sequences**. To identify which nucleotides and which positions are important in determining promoter strength, we calculated the odds ratio of nucleotide frequency between high-productive and low-productive promoter sequences. The odds ratio removes any bias coming from the random library generation (Supplementary Table 8). The most important bases contributing to promoter strength are those surrounding the Pribnow box (Fig. 3a), especially upstream of the Pribnow box (−17 to −13; Supplementary Fig. 4a, b). This is concordant with the structural observation that the −17 to −13 region comes into physical contact with the sigma factor[36]. Notably, C is significantly absent in this region (Supplementary Data 3). We found that as for other Gram-positive bacteria the presence of a TGN motif (extended Pribnow)[37] as long as the N is not a C (Fig. 3b). In agreement with a previous study of *M. pneumoniae* promoters[32, 38], we did not find a strong enrichment for nucleotides corresponding to the −35 box using canonical motif search methods (Supplementary Table 9). However, there is a slight sequence bias toward nucleotides present in the −35 sequence TTGCCA (−37 to −32) among the high-productive promoters (Fig. 3a). This sequence motif was weakly enriched in endogenous promoters of *M. pneumoniae*[39]. We also found that tandem Pribnow boxes increase transcriptional activity ($P < 10^{-10}$; two-tailed *t*-test (Log10DAMRatio of no tandem Pribnow cases vs. tandem cases), Supplementary Fig. 4c) and were able to determine different transcriptional strengths of alternative sequences, albeit the canonical TATAAT was still the optimal one (Fig. 3c and Supplementary Table 10).

To determine whether nucleotide combinations cooperatively contribute to promoter strength (epistasis), we used mutual

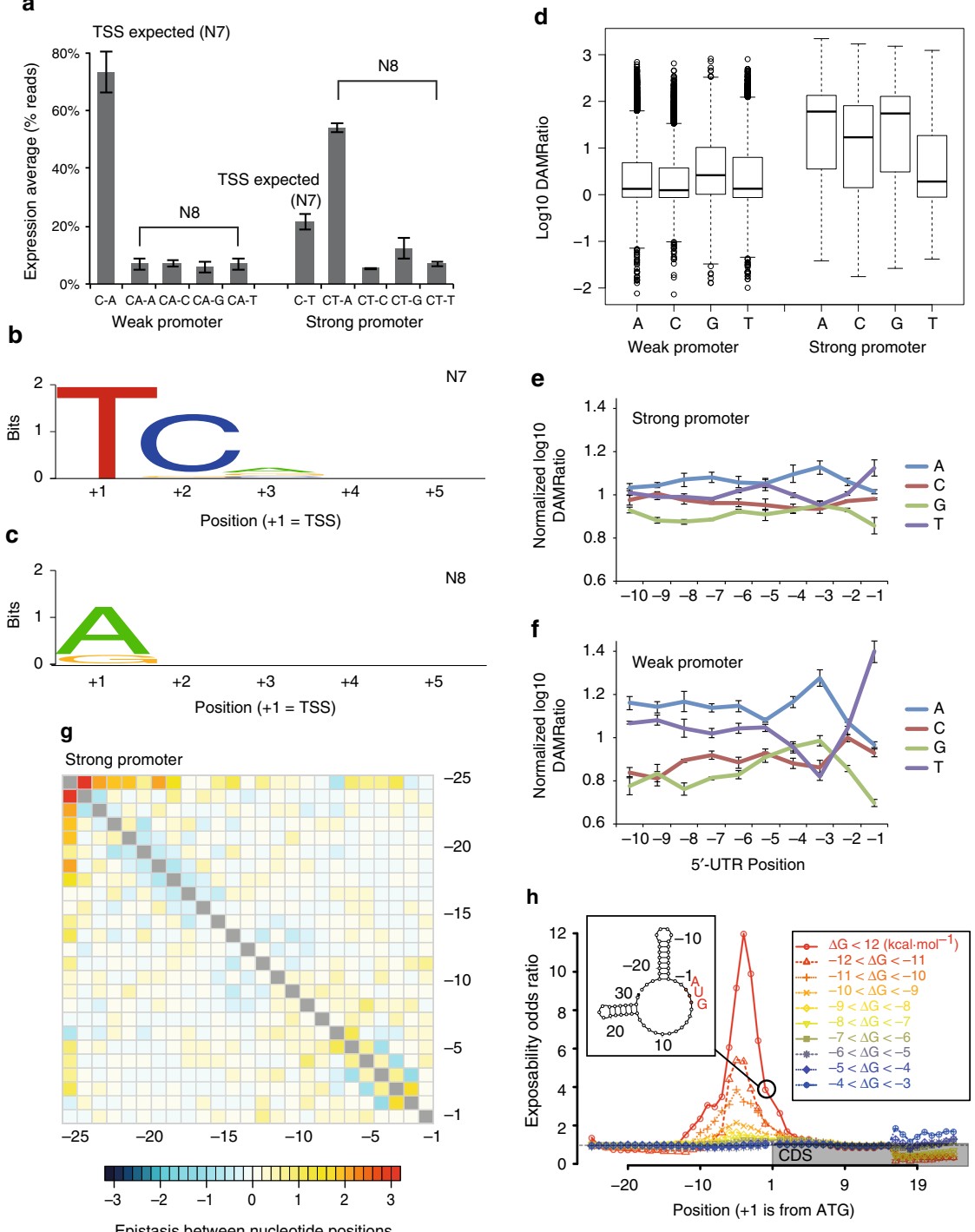

**Fig. 4** The effects of alternative transcription start sites and deep-sequence analysis of the 5′-UTR regions of *M. pneumoniae*. **a** Expression average according to real TSSs. *Error bars* represent standard deviation of two biological replicates. **b** Logo for sequences of N7 mRNAs with the strong promoter (*n* = 4637). **c** Logo for sequences of N8 mRNAs with the strong promoter (*n* = 9863) **b**, **c** We used N7 and N8 mRNAs commonly found in both RNA-seq experiments. The sequence logos were generated by WebLogo. **d** DAMRatio distributions according to the first randomized bases (theoretical +2). **e** The effect of specific nucleotides near the TSC on protein levels in the strong promoter set-up. DAMRatios were normalized within each subgroup starting with the same base. The first four positions upstream of the start codon are the key determinants of protein expression. **f** The effect of specific nucleotides near the TSC on protein levels in the weak promoter set-up. **e**, **f** *Error bars* represent standard deviation of four subgroups. **g** Epistatic interactions between nucleotide positions of the 5′-UTR in the case of the strong promoter. The first and second bases have strong epistatic interactions. **h** Investigation of structural effects near the TSC on protein levels for the strong promoter set-up. A secondary folding structure of mRNA is shown as an example. In this case, we considered that bases −22 to −3 and 15 to 29 are hidden (inside the structure) and that the others are exposed (outside the structure). We assessed whether a specific position is exposed or not, and then compared the level of exposure for each site in high-productive and low-productive sequences. To remove the folding energy effect, we binned sequences with $\Delta G = 1$ kcal mol$^{-1}$

information (MI) analysis[40]—the most general method to measure the dependency of two variables. We found that positions −17 to −13 before the Pribnow strongly interact with each other (up to a sevenfold stronger interaction than in random cases; Fig. 3d). Physical interaction between nucleotides and the sigma factor could explain these local epistatic interactions (Fig. 3e, f). Interestingly, both position −15 and, albeit to a lesser extent, position −16, are hubs of epistatic interactions. They connect with nucleotides downstream of the Pribnow box, and in particular with the TSS. We suspected these long-range interactions to be related to the selection of the TSS; specifically, we saw weak extended Pribnow signals when C/T are at the +1 (Supplementary Table 11). Moreover, we found moderate epistatic interactions between consecutive bases in the Pribnow–TSS spacer region, and weak interactions involving residues −20, −21, and −31 to −35. The overall level of epistasis, however, must not be very high considering that a naive Bayes classifier, which assumes that each position has an independent effect, enabled us to distinguish between the high-productive and low-productive promoters of our library (0.90 AUC; Supplementary Fig. 5).

**5′-UTR screening reveals functional leader sequences**. Translation efficiency in bacteria is related to the SD sequence and the RNA secondary structure of the 5′ of the mRNA and the first bases of the protein-coding region[32]. Recently, it has been reported that A and G are preferentially enriched at TSSs in some bacteria and that this could affect RNA levels[33, 41].

To determine the sequence contributions to translation efficiency, we introduced 25 randomized nt before *dam* under the control of two distinct promoters and applied ELM-seq and RNA sequencing (RNA-seq). It is known that TSSs in bacteria are preferentially found seven bases (at N7) downstream of the end of the Pribnow box motif (i.e., the theoretical +1 position). In the screens the +1 position was fixed, the strong promoter containing an unfavorable T and the weak promoter a favorable A (Fig. 2a). We identified the TSSs and the mRNA abundances using Dam-specific RNA-seq in two biological replicas (Supplementary Note 3, Supplementary Data 4, and Supplementary Fig. 6). The DAMRatios obtained from two biological replicas for the ELM-seq screen were significantly correlated (Supplementary Fig. 2c, d).

In the case of the strong promoter, we surprisingly found that, in two-thirds of the cases, the TSS actually shifted to the theoretical +2 position (N8), especially if it was an A or a G (Fig. 4a and Supplementary Data 4). When the theoretical +2 position was occupied by a C, the TSS remained at N7, the T (Fig. 4b). No other downstream preferences affecting these TSS shifts were found after the theoretical +2 position (Figs. 4b, c). On the other hand, as the weak promoter contains an A at the expected TSS, we found that the majority of transcripts began at the N7 position (Fig. 4a).

With respect to activity, we observed that the DAMRatio is systematically higher for the 5′-UTRs having an A or G as the first nucleotide in the sequence (Fig. 4d). We confirmed this first nucleotide bias in four individual 5′-UTR constructs (UTR1, 2, 7, and 8; Supplementary Table 12a and Supplementary Fig. 7). Upon mutating the first base, we found that having either an A or a G consistently resulted in greater protein yields. This phenomenon also held true for a leaderless construct (Supplementary Table 12b). Interestingly, we observed that the mRNA levels determined by qPCR were highly correlated with protein amount and activity ($r > 0.95$, Supplementary Table 12a), indicating that the first base affects the RNA production and/or degradation. In support of the later hypothesis, we found that the

actively translated constructs systematically had more RNA. In effect, when we mutated an internal TSC (mRNA sequence: 5′-ATC-ATG-dam) to any NTG combination, we found at the protein level the expected preference for the TSC codons (ATG = GTG > TTG >> CTG, Supplementary Table 13). At the same time, we found a strong correlation of protein and RNA levels, implying that RNA levels are dependent on their translation.

We then removed this first base effect and another confounding factor, the additional TSCs (Supplementary Note 4, Supplementary Fig. 8, Supplementary Tables 14 and 15) from the analysis by taking subgroups of sequences having the same TSS and no alternative TSC. The four subgroups (first base A, C, G, and T) show similar preferences, with stronger correlations when looking at the eight bases immediately upstream of the ATG ($r > 0.95$). Quite interestingly, we found that an A at −3 or a T at −1 consistently increase Dam activity (40% more compared with a T or a G at the respective positions) in all subgroups (Fig. 4d, e, Supplementary Table 16, and Supplementary Fig. 8d). We found that 5′-UTR sequences containing a SD sequence do not have higher Dam activity (Supplementary Fig. 9a, b). In fact, only a few ribosomal genes of polycistronic operons show a canonical SD[42]. However, when examining sequences of similar folding energies to control for the stability of mRNA secondary structure, we found that SD-like sequences at position −20 to −1 have a slightly, but significantly higher DAMRatio (Supplementary Table 17).

Finally, we used MI to quantify the epistasis between any couple of positions in the randomized 5′-UTR of the strong promoter (Fig. 4g) and weak—promoter translation screens (Supplementary Fig. 9d). We only found a strong epistatic interaction between the first and second nucleotide of the strong promoter, which can be explained by an unfavorable T at the theoretical +1 position, as we noticed previously.

To determine whether 5′-UTR folding stability affects translation in *M. pneumoniae*, we computed the correlation between mRNA folding energy and the corresponding DAMRatios of constructs that varied in length in the mRNA folding region[17] (Supplementary Fig. 9c). Similar to *E. coli*[17, 43], we observed that the mRNA folding energy of the 5′-UTR alone does not correlate well with Dam activity ($r < 0.1$). The correlation increases for both strong and weak promoters when the folding region contains up to ~30 nucleotides of the coding sequence (Supplementary Fig. 9c and Supplementary Data 5). Furthermore, we found the local hairpin structures near the upstream of ATG hamper translation (Fig. 4h).

Finally, we generated a predictor by incorporating the sequence determinant and folding energy information from our 5′-UTR libraries. Upon using only sequence features in the promoter analysis, we obtained moderate prediction power from the naive Bayes classifier (0.73 and 0.67 AUC for the strong and weak promoters, respectively), indicating that sequence alone does not determine translation efficiency. However, by combining both the mRNA folding energies and 5′-UTR sequence determinants, we were able to improve translation prediction (0.80 AUC; Supplementary Fig. 10).

## Discussion

Here, we developed a new, to the best of our knowledge, reporter assay called ELM-seq, and used it to simultaneously quantify the protein activity of hundreds of thousands of constructs by deep sequencing. This technique enables the in vivo identification of sequence functionally important determinants in bacterial promoters and 5′-UTRs. Our assay is an alternative to classic reporters and allows very high-throughput with no cell sorting or selection biases. It is a technically simple and robust method that

enables enrichment of both high-productive and low-productive constructs.

With ELM-seq, we not only were able to find the known elements and but also to quantitatively assess the effects of functional elements in *M. pneumoniae*. For example, we could determine the relative strengths of the extended TGN box motif (TGG = TGT > TGA > TGC) and that of the canonical Pribnow box motif (TATAAT > TAAAAT > TAGAAT > TACAAT). Also, we found that the nucleotide positions immediately upstream of the Pribnow box are the most important features for determining promoter strength when the Pribnow sequence is fixed. Interestingly, we found epistatic interactions within the promoter. There are three interpretations for these results. First, two or more proteins could bind to different regions of the promoter and have a cooperative interaction. Second, a protein with various domains (i.e., sigma 70) could bind in a cooperative manner to different regions of the promoter, and therefore non-adjacent sequences would have an epistatic effect. Third, a single domain binding to several bases would have a network of cooperative interactions (e.g., H bonds, hydrophobic interactions, conformational entropy, etc.), causing adjacent bases to have an epistatic interaction. In our case, considering the distance between epistatic interactions, the second and third scenarios are more likely.

For the 5′-UTR, we found that the SD motif is less important for translation initiation, as it had been foreseen from genome analysis[42]. This is at least true for monocistronic mRNAs (the way the screening was done), but could be more relevant for intra-operon cistrons, as seen from the higher frequency of SDs inside operons in vivo (~21 vs. ~35%).

Apart from individual contributions, we estimated the epistatic relationships of the promoter. The physical interaction network between the sigma factor and the DNA sequence upstream of the Pribnow motif introduces a local epistatic block. We also observed a long-range interaction between the upstream region of the Pribnow box and the base in the position of the expected TSS. Depending if this position was occupied by a purine or a pyrimidine, we saw a different preference for C at the −17 to −13 positions. Interestingly, these scenarios coincide with the fact that a weak sigma factor–polymerase interaction is required to shift the TSS when the open complex is disturbed[44].

Using the data generated from our screens, we developed a fairly accurate predictor for promoter strength and translation efficiency that can be used for the engineering of robust gene circuits in *M. pneumoniae*. We found our prediction method to be accurate after training with only ~2000 sequences—a very small number considering that all the possible sequence spaces have more than hundreds of trillions of combinations. This indicates that our screen can be done on a small scale. With the current sequencing power, we could obtain a comprehensive list of promoter and 5′-UTR sequences with a good dynamic range for transcription and translation. These sequences could then be directly used for synthetic design.

Our method can be easily applied to any bacterium that does not have Dam activity, or restriction sites overlapping with GATC or eukaryote (where DamID has been extensively used)[45]. This method could be combined with the knockdown of transcription and translation factors, and with different perturbations, thereby allowing the identification of more detailed sequence contributions to gene expression.

## Methods

**Cell culture**. *E. coli* chemically competent cells were either DH5α (NEB) or TOP10 (Invitrogen). High-competent electrochemical competent cells MegaX DH10B T1 (Invitrogen) were used for the random library plasmid transformation. *M. pneumoniae* M129 (passage 34) was grown in modified Hayflick medium in

150 cm² flasks and transformed by electroporation as previously described[46]. Briefly, cells were split 1:10, and washed and collected in 300 µl Electroporation buffer (8 mM HEPES·HCl, 272 mM sucrose, pH 7.4) 3 days later. Fifty-microliter cells were used to electroporate 5 µg plasmid (1 mm gapped cuvettes, 1.25 kV, 100 Ω, 25 µF, in a Gene Pulser Xcel Electroporator, Bio-Rad). Cells were recovered in Hayflick at 37 °C for 2 h and diluted 1:5 in Hayflick supplemented with 80 µg ml⁻¹ Gentamycin.

**Cloning**. *Dam* was PCR-amplified from *E. coli* genomic DNA using F_dam_Acc65 and R_dam_Nsi oligos (see Supplementary Table 1 and Supplementary Data 1), and introduced into a pGEM-T-Easy vector (Promega) by AT-cloning following the manufacturer's instructions. It was then subcloned into the minitransposon plasmid pMT85 (modified from Professor Richard Herrmann, see Supplementary Table 1) with Acc65I and NsiI sites (the internal NsiI site in *dam* had previously been mutated in the pGEM vector by site-directed mutagenesis using Pfu, Life Technologies) in frame with a C-terminal Flag tag (see below). Promoters for the proof of principle were PCR-amplified or ordered as linkers (all oligonucleotides cited in the Methods are in Supplementary Data 1) and cloned with NotI and Acc65I upstream of *dam* (see Supplementary Table 1). Afterwards, 4× GATC sites were included just before the promoter in a linker flanked with NotI and EagI sites (oligos: F_GATC4, R_GATC4). The PCR products and digestions were checked on 1% agarose (Sigma)/TAE buffer (40 mM Tris·Acetic, 1 mM EDTA, pH 8.3) gels and stained with Gel Red (Biotium). Bands were purified using the Gel Extraction Kit (Qiagen).

In the case of the screening constructs, we used the Gibson assembl (GA)y strategy to join multiple DNA fragments in a single, isothermal reaction. Fragments were obtained either from annealed oligos (linkers) or PCR with overlapping regions between each other and the NotI–EcoNI cut plasmid (see Supplementary Note 1 and Supplementary Table 1). The promoter screen linkers were obtained by annealing a 50 µM mix of the two overlapping oligos, F_pMT_4GATC and R_pMT_4GATC in water, and heating during 2 min at 95 °C and then 10 min at 52 °C; the *dam* containing fragment was obtained by PCR of the pGEM-dam using F_Prom_Dam and R_Dam_pMT oligos. In the case of the translation screen, linkers were obtained by primer annealing (as above) and extension using Klenow polymerase (*exo*-, NEB). The primers used were F_pMT_SyP32 +R_SyP32_Dam and F_pMT_llmp20 +R_llmp200_Dam and the *dam*, common to both constructs, was amplified with F_Dam and R_Dam_pMT oligos (see above). The GA was performed by mixing 50 ng of vector with equimolar amounts of the fragments. GA enzyme mix was prepared at the CRG Protein Expression Service according to the original paper[47] with NEB reagents, except for the *Taq* ligase that was produced recombinant in house. Then, the mixture was heated at 50 °C during 50 min. Three 20-µl reactions were pooled in order to obtain a large enough representation of random sequences, ethanol-precipitated, and then electroporated into DH10B T1 (Invitrogen) according to the manufacturer's instructions. Fifteen 150-mm plates were used for each construct and selected with kanamycin. In summary, in the transcription screen, the promoter sequence consisted of a constant −10 box (a standard Pribnow box, sequence TATAAT) surrounded by 25 random nucleotides (N25) upstream and seven downstream (N7). We included a −10 box to control for the site of transcription initiation. For the translation screens, we fused 5′-UTR sequences composed of 25 random nucleotides to both a strong and a weak promoter (Fig. 2a).

The validation (promoter and 5′-UTR) constructs were cloned similarly to the screening constructs, and the first base mutations were introduced with a C-terminal Flag-tagged *dam* to enable detection by western blot analysis (see Supplementary Table 1).

Integrity of all the constructs was checked by Sanger sequencing (at GATC Biotech).

**Transformation and genomic DNA isolation**. Cells were transformed by electroporation and selected with gentamycin. The Tn4001 minitransposon vector allows to insert libraries into the genome, with only one individual sequence being incorporated per genome in ~95% of the cases (Raúl Burgos, personal communication). For genomic DNA purification (modified from ref. [46]), cells were inoculated in a 75-cm² flask (1:10 dilution) and grown for 3 days in Hayflick. Then, cells were washed once with phosphate-buffered saline (PBS) and scraped in the same buffer. After collection by centrifugation, 300 µl of lysis buffer (50 mM Tris·HCl, 25 mM EDTA, pH 8) at room temperature (RT) was directly added to the pellet, and 25 µg ml⁻¹ RNAse A was added after resuspension. This was followed by an incubation at 37 °C during 30 min. Then, SDS to 1%, plus proteinase K to 0.40 mg ml⁻¹ were added, mixed, and incubated for 2 h at 55 °C. The reaction was cooled at RT for ~5 min. Phase lock Eppendorf tubes (MaXtract low density, Qiagen) were centrifuged at 1500 × *g* during 5 min at RT. The extraction was conducted by adding 1 volume of phenol–chloroform to the lysate, mixing gently (no vortex), and spinning at 1500 g for 5 min. The aqueous (upper) phase was taken and the process was repeated with chloroform. The last upper phase was transferred into a clean tube, where 2.5 µl of 20 mg ml⁻¹ glycogen and 1/10 volume of 3 M NaOAc, pH 5.5 had been added. Finally, DNA was precipitated with 2.5 volumes ethanol and resuspended in 100 µl TE (10 mM Tris·HCl, 1 mM EDTA, pH 7.5).

**Dam activity detection by qPCR.** For the proof of principle, we used promoters previously characterized in our laboratory that result in different protein amounts. After transformation of *M. pneumoniae*, a sample was prepared for protein extraction (see following section) and another one for genomic DNA prep, digestion, and qPCR. In the case of the activity assay, 1 µg of genomic DNA (see previous section) was digested in 20 µl either with 10 U DpnI in NEB buffer 4 or with 5 U MboI (DpnII isoschizomer) in MBI-Fermentas buffer R at 37 °C overnight (O/N). Inactivation was carried out during 20 min at 80 °C. Digestion was confirmed in a 0.7% agarose gel (see Supplementary Fig. 1). DNA was diluted and 5 ng were used for each qPCR (GoTaq qPCR Master Mix, Promega) reaction with oligos that are external to the 4× GATC cassette (F_uGATC2 + R_q674) or certain genomic GATC loci (oligos R_19580.1 + F_19580.2, R_19580.2 + F_170908.1 + R_170908.1, F_170908.2 + R_170908.2, see Supplementary Data 1). rRNA 16S oligos (F_q16S + R_q16S) were used as a reference (amplicon without any GATC), and each digestion was normalized independently with the 16S qPCR.

**Western blot.** For protein detection, cells grown for 3 days in a 25 cm$^2$ flask were washed with PBS and extracted with 150 µl 1% SDS in TE prior to protein quantification by bicinchoninic acid (BCA) (Pierce). Fifty micrograms of total protein were loaded per gel, and a western blot was performed using M2 anti-Flag mouse monoclonal antibody (Sigma, Cat No. F3165, 1:2000 dilution) or anti-GFP monoclonal antibody (Roche, Cat No. 11814460001, 1:2000 dilution) and detected with anti-mouse IgG conjugated to horseradish peroxidase (Jackson, Cat No. 515035003, 1:10,000 dilution). Blots were developed using high-sensitivity electrochemiluminescence (ECL) reagent (Thermo Fisher Scientific) and visualized using the Fujifilm LAS-3000 developer. Intensities of immunoreactive bands on western blots were quantified using Quantity One (Bio-Rad) software. Protein levels were normalized using a ribosomal protein (RL7) antibody as a loading control[30] (1:500 dilution).

**mRNA quantification by RT-qPCR.** RNA was extracted from cells growing in exponential phase (24 h post inoculum) with Qiazol and purified according to the manufacturer's instructions (miRNeasy kit, Qiagen), including an in-column DNase digestion. RNA concentration was measured in a Nanodrop UV-Vis Spectrophotometer (Thermo Fisher Scientific). A standard RT (retrotranscription) reaction was carried out, in which random hexamers (37.5 ng µl$^{-1}$ final, Invitrogen) were hybridized with 1 µg of total RNA, during 2 min at 95 °C, and then at 65 °C during 5 min before placing it on ice. After hybridization was performed, the RT reaction was set up by adding 4 µl of 5× buffer, 2 µl of dithiothreitol (DTT) at 0.1 M, 1 µl of dNTPs at 10 mM, 1 µl of SCII (Invitrogen), and 1 µl RNasin 40 u µl$^{-1}$ (Promega) directly to the primed RNA (20 µl final reaction volume). The mix was incubated at RT for 10 min and at 42 °C for 50 min, and then heat-inactivated (70 °C, 10 min). qPCR was done with the 2× GoTaq qPCR Master mix (Promega), 0.15 µM oligos, and 0.5 µl cDNA in a Light Cycler-480 (Roche), in standard conditions, a 384-well format and in at least triplicates. Two sets of *dam*-specific oligos were employed (F_qdam + R_qdam or F_qdam2 + R_qdam2) and MPN517 was used as a reference gene (F_q517 + R_q517).

**Quantification of Dam protein by tandem mass spectrometry.** Cells were grown in a 25-cm$^2$ flask for 3 days as above, washed with PBS, and lysed/collected in 4% SDS, and 0.1 M HEPES·HCl pH 7.5. Samples were reduced with DTT (15 µM, 30 min, 56 °C), alkylated in the dark with iodoacetamide (180 nmol, 30 min, 25 °C), and digested with 3 µg LysC (Wako) O/N at 37 °C and then with 3 µg of trypsin (Promega) for 8 h at 37 °C following FASP procedure (Filter-aided sample preparation[48]). After digestion, the peptide mix was acidified with formic acid and desalted with a MicroSpin C18 column (The Nest Group Inc) prior to LC-MS/MS analysis.

The peptide mixes were analyzed using a LTQ-Orbitrap Velos Pro mass spectrometer (Thermo Fisher Scientific) coupled to an EasyLC (Thermo Fisher Scientific). Peptides were loaded onto the 2-cm Nano Trap column with an inner diameter of 100 µm packed with C18 particles of 5 µm particle size (Thermo Fisher Scientific) and were separated by reversed-phase chromatography using a 25-cm column with an inner diameter of 75 µm, packed with 1.9 µm C18 particles (Nikkyo Technos). Chromatographic gradients started at 93% buffer A and 7% buffer B with a flow rate of 250 nl min$^{-1}$ for 5 min and gradually increased 65% buffer A and 35% buffer B in 120 min. After each analysis, the column was washed for 15 min with 10% buffer A and 90% buffer B. Buffer A: 0.1% formic acid in water. Buffer B: 0.1% formic acid in acetonitrile.

The mass spectrometer was operated in data dependent acquisition (DDA) mode, and full MS scans with 1 µ scans at a resolution of 60,000 were used over a mass range of *m/z* 350–2000 with detection in the Orbitrap. Auto gain control (AGC) was set to 1e$^6$, dynamic exclusion (60 s) and charge state filtering disqualifying singly charged peptides were activated. In each cycle of DDA analysis, following each survey scan the top 20 most intense ions with multiple charged ions above a threshold ion count of 5000 were selected for fragmentation at normalized collision energy of 35%. Fragment ion spectra produced via collision-induced dissociation were acquired in the Ion Trap, AGC was set to 5e$^4$, isolation window was 2 *m/z*, activation time was 0.1 ms, and maximum injection time of 100 ms was used. All data were acquired with Xcalibur software v2.2.

**Growth curves.** In order to obtain equal amounts of each sample, initial inocula for the growth curves were quantified. Briefly, cells were grown for 3 days in a 25-cm$^2$ flask, collected in 1 ml medium, and 100 µl was used for quantification with a BCA protein assay kit (Pierce). Same amounts of total protein (1 µg) were aliquoted per well in a 96-multiwell plate in duplicates. Two hundred microliters of Hayflick medium were added per well, and the cells were incubated in a Tecan Infinite plate reader at 37 °C. Growth index (absorbance 430/560 nm, settle time at 300 ms, and number of flashes equal to 25) was taken every hour for 5 days as published[46]. To quantify growth, we determined two slopes in the growth curve. The first one is based on the time interval from 10 to 30 h (early slope) and the second one on the whole growth curve (late). The early slope was determined by considering the maximum median of the slope between two time points (Eq. 1) separated by three time measurements over successive periods of 30 time points. The late slope was determined by considering the maximum median value of the slope between two time points separated by four time measurements (Eq. 2) over successive periods of 30 time points.

$$\text{Early slope} = (\text{value}(\text{time}[i]) - \text{value}(\text{time}[i+3]))/(\text{time}[i] - \text{time}[i+3]) \quad (1)$$

$$\text{Late slope} = (\text{value}(\text{time}[i]) - \text{value}(\text{time}[i+4]))/(\text{time}[i] - \text{time}[i+4]) \quad (2)$$

The early slope is more representative of growth, while the late slope reflects the metabolic activity.

**Screen-sequencing library prep.** Genomic DNA was prepared and digested as above (see "Dam activity detection by qPCR"). For the library prep, 50 µl reactions were prepared with 0.5 µM oligos and 50 ng of digested input DNA. Using Phusion polymerase (NEB) DpnI-treated DNA was amplified during 12 cycles, whereas MboI-treated DNA had to be amplified during 15 cycles with the oligos F_SE_i1_dam to F_SE_i9_dam (nine indexes) and R_SE_Tn (Supplementary Data 1 and Supplementary Note 1). PCR products were purified with 50 µl AMPure XP beads according to the manufacturer's instructions (Agentcourt), and DNA resuspended in 22 µl of Elution Buffer (10 mM Tris·HCl, pH 7.5, EB). Libraries were quantified using the Illumina KAPA quantification kit according to the manufacturer's instructions (Kapa Biosystems).

**Gene-specific RNA-seq.** A modification of SHAPE-seq[49, 50] was developed to sequence only *dam* mRNAs. As a first step, RNA was extracted as above (see mRNA quantification by RT-qPCR), measured with a Nanodrop, and its integrity confirmed in a 6000 Nano chip Bioanalyzer (Agilent).

Two approaches were optimized (see also Supplementary Note 1).

First approach: In order to obtain the mRNA from our *dam* construct, we amplified cDNA using a specific RT-PCR. *dam*-specific primer (RT_r2dam2) at 2.5 µM was hybridized to 5 µg of total RNA and the RT reaction was carried out as above (see "mRNA quantification by RT-qPCR").

Once the reaction was finished, RNA was removed by first adding 1 µl of 4 N NaOH during 5 min at 95 °C, and then by neutralizing it with 2 µl of 1 M Tris·HCl (pH 8). Then, we added 1 µl of RNase cocktail (Ambion) and incubated the tube at 37 °C during 30 min. After this, the reaction was brought to 50 µl with $H_2O$ and the ssDNA was cleaned with 1 volume of AMPure beads as above. Finally, we eluted with 25 µl of EB.

We also performed an enrichment step, in which 1 µl of Biot_dam oligo (10 µM) was added with 0.5 µl of EDTA (200 mM) in 20 µl 2 × SSC (saline-sodium citrate) and heated at 96 °C for 5 min and then at 55 °C during 5 min. After this, we resuspended 25 µl of Streptavidin magnetic bead suspension (M-280, Invitrogen) in 20 µl of 0.5 × SSC and added the DNA to the mixture. To enable capture of the biotinylated oligo, the previous mixture was incubated for 10 min at RT and then separated with the magnetic holder during 30 s. The precipitate was washed four times with 200 µl of 0.1 × SSC at RT, and, afterwards, the product was released from the biotinylated oligo by incubating the particles twice with 30 µl of $H_2O$ at 70 °C for 5 min and then immediately putting in the magnetic holder.

After finishing the enrichment process, the ssDNA ligation was performed. We prepared a 50 µl reaction with 5 µl of 10× buffer, 1 µl of ATP, 2.5 µl of MnCl$_2$, 2 µl of CircLigase (Epicenter), 0.84 µl of 100 µM linker (L_read1), and 15 µl ssDNA (from the previous step). This mix was incubated at 65 °C during 120 min, and then the ligase inactivated at 85 °C during 15 min. The reaction was cleaned with 1 volume AMPure beads as above and eluted with 25 µl of EB.

The final step involved a PCR reaction in order to introduce Illumina sequences and thus prepare the libraries for sequencing. This reaction was performed with 4 µl of 5× buffer, 0.4 µl of dNTPs, 0.4 µl of Phusion HF polymerase (NEB), 0.5 µM of primers (F_PEu and indexed R_PE), and 5 µl of ssDNA in a 20 µl final reaction volume. We PCR-amplified during 20 cycles (elongation at 60 °C). Each PCR product was checked in a 2% agarose/TAE gel, and once the correct band was obtained (~250 bp), we repeated the same reaction but in a 50 µl final volume. The PCR was cleaned and selected by size using AMPure beads (as above), and quantified by qPCR for the ultrasequencing process.

Second approach: we obtained the cDNA at standard RT, as above (mRNA quantification). Once the reaction was finished, RNA was removed and cleaned as before (first approach).

In this Second approach, the biotin enrichment step was not needed to get a clean PCR product, and the ligation was performed directly on the clean ssDNA from the former step. We prepared a 50 μl ssDNA ligation reaction as in the first approach. The reaction was cleaned with AMPure beads as above and eluted with 25 μl of EB.

In the last step—the PCR reaction—we specifically amplified the *dam* transcripts. The reaction was performed with 10 μl of 5× buffer, 1 μl of dNTPs, 1 μl of Phusion HF polymerase (NEB), 0.5 μM of a general forward (F_PEu), and a *dam*-specific reverse primer (R_PEi6_dam2 or R_PEi12_dam2, depending on the index) and 10 μl of ssDNA in a 50 μl final reaction volume. Samples with a high signal (i.e., strong promoter) were amplified during 20 cycles (a 60–70 °C annealing gradient was introduced for the first 10 cycles), while samples with a weak signal (weak promoter) were amplified for 22 cycles. PCR products were checked on a 2% agarose gel, and, once we obtained the correct band, we performed the same reaction but in a 50 μl final volume. Then, the product was cleaned and selected by size with 1 volume AMPure beads as above, and quantified by qPCR for the ultrasequencing process (with KAPA kit as above).

**Ultrasequencing**. A sample of all indexed libraries was prepared at 4 or 10 nM in 10 μl $H_2O$. All libraries were subjected to quality control using a Bioanalyzer High Sensitivity DNA Assay chip (Agilent). double-stranded DNA samples were cluster-amplified and sequenced in the HiSeq 2500 platform (Illumina) at the CRG Genomics Core facility.

**Proteomics data analysis**. Proteome Discoverer software suite (v2.0, Thermo Fisher Scientific) and the Mascot search engine (v2.5, Matrix Science[51]) were used for peptide identification. Samples were searched against a *M. pneumoniae* database[52] with a list of common contaminants and all the corresponding decoy entries (87,059 entries). Trypsin was chosen as enzyme and a maximum of three miscleavages were allowed. Carbamidomethylation (C) was set as a fixed modification, whereas oxidation (M) and acetylation (N terminal) were used as variable modifications. Searches were performed using a peptide tolerance of 7 ppm, a product ion tolerance of 0.5 Da. Resulting data files were filtered for false discovery rate (FDR) < 5%. Protein Top 3 areas were calculated with unique peptides per protein.

**Analysis and filtering of Illumina sequencing data**. Raw reads were filtered for both the common Dam-coding sequence and the varying upstream sequence according to the specific study. Since the raw reads are reverse complements, we used the following reverse complementary sequences for filtering:

1. Dam coding sequence: TGCCCACTTCAAAAAAGCGCGATTTTTCTTCAT.
2. Promoter study: GACCGGAACTTCTATGATCGAGATCGAGATCGA-GATCGCGGCCGCAAC and the TATAAT motif in the middle of the promoter sequence.
3. 5′-UTR study with strong promoter: AGTTTATATTATAACACTTTAACC-TATGGC.
4. 5′-UTR study with weak promoter: TGCAATTATTCTAACAAACCC CAAACTTATTTCAA.

For each study, we used three different indexes to label three different experimental conditions. No cut, DpnI-treated, and MboI-treated samples were separately labeled with Illumina index. After filtering the raw reads with exact matches, we counted the read counts for each variant and normalized the raw read counts by the total read counts to obtain a counts per million (CPM) value.

$$\text{CPM}_i = \frac{X_i + 1}{N + 1} \times 10^6$$

Where $X_i$ is the number of reads for a certain sequence and $N$ the number of total reads in one experimental condition.

$$\text{DAMRatio} = \frac{\text{CPM}_i}{\text{CPM}_j}$$

DAMRatio is calculated by dividing $\text{CPM}_i$ from the MboI cut by the $\text{CPM}_j$ from the DpnI cut experiments.

In order to be confident in our results, we only used reads that were found more than 100 times in at least one of the digestions (see Supplementary Table 4). From the two biological replicates of the three screens we obtained 49,706 and 56,599 random promoter sequences, 51,746 and 57,781 5′-UTR sequences with the strong promoter, and 53,886 and 34,121 5′-UTR sequences with the weak promoter. Of these sequences, 23,666, 14,582, and 16,529 were common between replicates and further used for correlation analyses (Supplementary Fig. 2). From the linear regression model, we chose parameters in order to compute combined DAMRatio. To compare the log-transformed DAMRatios among different experiments, we standardized them by fitting a mixture of Gaussian distributions and setting the left-most peak as zero (see DAMRatio distribution fitting).

**Motif discovery**. To identify enriched motifs, we used the motif discovery algorithm EXTREME[53] with its default settings (maximum gaps of 10 bases and minimum motif length of three bases). In the transcription and translation screens, we used input sequences belonging to the high-productive type as the positive set and belonging to low-productive type as the negative set.

**MI analysis**. We quantified the epistasis of two nucleotide positions by using MI—the most general method for measuring the dependency of two variables[40]. For each pair of nucleotide positions $i$ and $j$, we computed the MI between the number of all bases of $i$ and $j$ with high-productive promoter sequences (MI(observed)). To remove the intrinsic relationship between nucleotide positions derived from library generation, we randomly picked the same number of sequences and generated a basal MI level (MI(random)). An epistatic interaction was defined by the ratio of MI(observed) over MI(random). We used the dminjk. pw function of the "mpmi" package in R (version 2.7), which calculates the MI between a set of discrete variables (nucleotides) between two vectors. To eliminate random selection bias, we averaged out 10,000 random selections of 10,000 samples.

**Naive Bayes classification and support vector prediction**. The naive Bayes classifier is a well-known statistical learning method based on Bayes' theorem. It is based on the very simple assumption that all feature variables are independent. Despite its simplicity, Bayes classifier has been successfully used to study promoter recognition[54]. We used naive Bayes classifier to evaluate the independent contributions of individual bases in promoter sequences. This method requires vectors of real numbers in feature space. To convert a sequence into vectors of feature space, we coded the input data set as a binary, orthonormal set of four dimensions. Each nucleotide was set as (1, 0, 0, 0), (0,1,0, 0), (0, 0, 1, 0), and (0, 0, 0, 1), for 'A', 'C', 'G', and 'T', respectively. We divided the sequences into two classes based on the probability of Gaussian fitting. If sequences had more probability of belonging to high-productive sequences than to low-productive sequences, we set them as positive sets; otherwise, we set them as negative sets.

The support vector machine has also been successfully used for promoter identification[55]. Since it implicitly considers the hidden interactions between variables, we used it to make a final prediction model. Similar to the naive Bayes classifier, it requires vectors of real numbers in feature space. As such, we converted the sequence into vectors of feature space as (1, 0, 0, 0), (0,1,0, 0), (0, 0, 1, 0), and (0, 0, 0, 1), for 'A', 'C', 'G', and 'T', respectively. With these converted vectors, we used support vector regression (SVR) to train the model to predict continuous values (DAMRatio). Basically, the SVR solves this:

$$\text{Minimize} \quad \frac{1}{2}\|w\|^2 + C\sum_{i=1}^{n}\left(\xi_i + \xi_i^*\right)$$

$$\text{subject to} \quad \begin{cases} y_i - \langle w, x_i \rangle - b \le \varepsilon + \xi_i \\ \langle w, x_i \rangle + b - y_i \le \varepsilon + \xi_i^* \\ \xi_i, \xi_i^* \ge 0 \end{cases}$$

where $x_i$ is an input data sample and $y_i$ is the $\log_{10}$-transformed DAMRatio value. The inner product plus intercept $\langle w, x_i \rangle + b$ is the prediction for that sample, and $\varepsilon$ is a free parameter that serves as a boundary threshold (we set it as 0.2). $\xi_i$ and $\xi_i^*$ in the formula are non-negative slack variables that allow a certain violation of boundary error. $C$ is the regularization term that balances the training error. We used SVR function "sklearn" package with default parameters and radial basis function (rbf) kernel to transform input data to kernel space. To check whether the prediction is biased toward a given training set, we trained a support vector machine with nonoverlapping training sets (5000 sequences) and predicted an independent testing set (2000 sequences). The promoter prediction resulted in both different training sets being similar to each other (Pearson's correlation coefficient $r = 0.98$; $P < 10^{-10}$; Supplementary Fig. 5)

In fact, only a small number of sequences (about 2000 out of $2 \times 10^{19}$ result in an AUC of 0.90; Supplementary Fig. 5b–d) were needed to obtain a promoter predictor that was more accurate than one from a previous analysis[32] trained with natural *M. pneumoniae* promoters (0.70 AUC; Supplementary Fig. 5a). We improved the prediction of endogenous promoters of *M. pneumoniae* by being able to distinguish real promoters from non-promoter sequences that contain a Pribnow motif[32] (Supplementary Fig. 5e).

**Translation efficiency prediction with 5′-UTR folding energy**. To examine the structural features that universally affect translation efficiency, the folding free energies of each region were calculated by NUPACK[56] and correlated with the expression level in the case of each variant. We varied the mRNA folding start position (from −26 to −1, which denotes the 5′ end of the 5′-UTR, while +1 denotes the A of the translation initiation codon) and end position (from −10 to

100) to determine which folding regions are important in regulating mRNA translation. In total, we obtained 2014 × 94,945 + 2014 × 71,478 folding-expression correlation relationships. After conducting the same procedure on both strong and weak promoter translation screens, we compared their folding-expression correlation matrices and found that they were significantly similar to each other. In order to check how local hairpin structures affect translation (Fig. 4h), we used the folding region (−25 to 30) that gives maximum correlation between folding energy and DAMRatios. We assessed the secondary structures from the NUPACK output and calculated whether or not each position was located inside secondary structures.

In order to build a prediction model for 5′-UTRs, we used the same SVR and coding (binary, orthonormal set of four dimensions) as for the promoters to convert sequences into numerical vectors. Furthermore, we used the predicted folding energy ($\Delta G$) as an input with the sliding window method (up to 30 bases in coding regions). We used the regions having a correlation between $\Delta G$ and DAMRatio greater than 0 in order to remove the insignificant features, thereby reducing the problem of having too many input data features[57].

We examined whether translation efficiency prediction tools commonly used and optimized for *Escherichia coli* can be applicable for *M. pneumoniae*. We evaluated the prediction power of two predictors—the RBS Calculator[18] and the UTRDesigner[17]—and obtained an AUC of 0.66 for both of them. We used the default parameters and 5′-UTR and coding region sequences required for these predictors (Supplementary Fig. 10 and Supplementary Data 6).

**Gene-specific RNA-seq data analysis**. Gene-specific RNA-seq was used to determine the TSS of the Dam translation screen libraries. The raw sequencing reads were filtered for the *dam*-coding sequence (ATGAAGAAAAATCG). The resulting reads were aligned with the final confident 5′-UTR library sequences (94,945 5′-UTR sequences with the strong promoter and 71,478 5′-UTR sequences with the weak promoter) after trimming six barcode bases from their 5′ end (see "Gene-specific RNA-seq" and Supplementary Note 1). If the length of the 5′-UTR was 26 nts, then we only used the sequences correctly mapped to promoter +1 sequence (strong promoter with T and weak promoter with A). To remove mapping ambiguity, we discarded 5′-UTR lengths shorter than 15 bases. Finally, the number of reads mapping to a certain length of 5′-UTR was normalized by the total number of filtered reads using the following equation:

$$\text{TPM}_{i,l} = \frac{N_{i,l}}{N_{\text{tot}}} \times 10^6$$

Where $N_{i,l}$ is the number of the reads for a specific 5′-UTR length ($l$) of certain sequence ($i$). $N_{\text{tot}}$ is the number of total reads after filtering and TPM is transcripts per million.

From this, we estimated the TSSs by the 5′-UTR length with the maximum number of read counts. With these estimated TSSs, overall base preference near the TSS of N7 (26 nt long UTRs) and N8 (25 nt) *dam* mRNAs were calculated (Supplementary Table 18). For RNA abundance, we summed up $\text{TPM}_{i,l}$ from all the different lengths of 5′-UTRs mapped to a specific construct. To normalize the RNA abundances with DNA abundance, we used the log10-transformed DNA read counts from the "no restriction enzyme" treatment ("uncut" DNA data).

**Statistics**. Otherwise specified, we tested the statistical significance of the correlation coefficient $r$, using the stats.pearsonr function of the "scipy" python package. It converts a Pearson's correlation coefficient to an appropriate $t$-value, and statistical significance (two-tailed $P$ value) is then calculated using approximation of Student's $t$-distribution with degrees of freedom $n-2$. We also used Spearman rank-order correlation coefficient with stats.spearmanr function of the same package. It gives two-tailed $P$ value for null hypothesis that two sets of data are uncorrelated. For the $t$-test, we used the two-tailed $P$ value from the stats.ttest_ind function of the same package. As for the Fisher test, we used the stats.fisher_exact function in "scipy" package. AUC of receiver operating characteristic curves was calculated using the metrics.auc function of the "sklearn" python package. To correct the multiple testing errors, we used Bonferroni-corrected $P$ values calculated with a python script to multiply the obtained $P$ value by the number of testing trials.

**Code availability**. Computer code is available at https://github.com/lionking0000/ELMSeq/.

**Data availability**. The raw data of DNA sequencing of random promoter and 5′-UTR libraries, as well as of RNA-seq of the 5′-UTR libraries and the UTR pool internal control, have been submitted to the ArrayExpress short read database (http://www.ebi.ac.uk/arrayexpress) and assigned the identifiers E-MTAB-5365 and E-MTAB-5363, respectively. Proteomics data have been submitted to ProteomeXchange via the PRIDE database (http://www.ebi.ac.uk/pride) and assigned the identifier PXD005606.

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

## Acknowledgements

We acknowledge the staff of the Genomics core facility for their assistance, especially to Jochen Hecht for fruitful discussions on the screening designs. Also, we thank the CRG/UPF Proteomics Unit that is part of the *Plataforma de Recursos Biomoleculares y Bioinformáticos* (*Instituto de Salud Carlos III*), supported by grant PT13/0001. The project was supported by funds from the *Fundación Marcelino Botin* and the Spanish *Ministerio de Economía y Competitividad* (BIO2007-61762). This project was financed by *Instituto de Salud Carlos III* and co-financed by *Federación Española de Enfermedades Raras* under grant agreement PI10/01702 and the European Research Council (ERC) under the European Union's Horizon 2020 research and innovation program, under grant agreement Nos 634942 (MycoSynVac) and 670216 (MYCOCHASSIS). We acknowledge support from the Spanish Ministry of Economy and Competitiveness, *Centro de Excelencia Severo Ochoa 2013-2017* and the support given by *Juan de la Cierva-Incorporación* Program (IJCI-2014-22070) to J.S.Y. We also acknowledge the support of the CERCA Program/*Generalitat de Catalunya*. We thank Lambert Montava for initial clonings, and Verónica Llorens-Rico and Dr Guillaume Filion for comments on the manuscript. We appreciate all the feedback from the Serrano lab members.

## Author contributions

Conceived the project: E.Y., J.S.Y. Designed and performed the experiments: E.Y., A.S. Analyzed the data: J.S.Y. Wrote the paper: J.S.Y., E.Y., L.S. Supervised: L.S.

## Additional information

**Competing interests:** A patent has been applied for (GB 1708679.4) by L.S., J.S.Y., and E. Y. The remaining author declares no competing financial interests.

