## [Peer Review file · Nature Communications]

REVIEWERS' COMMENTS:

Reviewer #1 (Remarks to the Author):

Outstanding manuscript by Yus et al. on a new experimental and computational technique to quantify gene expression *in vivo*. It could be applied to many organisms and the authors validated it on an idiosyncratic organism such as *M. genitalium*, where the signal sequences for the initiation of transcription and translation are non-standard. The method relies on the *in vivo* expression of a DNA methylation enzyme followed by deep sequencing and data analysis using probabilistic and information theory models. The forward engineering of regulatory sequences is one of the limiting step in synthetic biology, where there will be a shift from characterizing existing biological parts (such as promoters and RBSs) to their *de novo* design. Many labs currently use more often synthetic RBSs libraries (obtained through computational design) than available known RBS. The reason for this paradigm shift is the lack of modularity of such parts and the dependency on the context. The Yus et al. manuscript uses a simple experimental technique to produce predictive models for both transcription and translation initiation. The data, Figures, Tables and text are of excellent quality. I recommend to publish the manuscript in Nature Communications after replying to the minor questions suggested below.

Questions:

- Intermediate levels of dam expression will create hemimethylations that will be counted as methylated, therefore overestimating the dam expression. Is this true? Of course, this systematic bias could be accounted in modelling. The discussion may benefit from a clarification on this.
- In principle, here any sequence region that would be a binding region for more than one protein would produce epistasis between all the nucleotides within the binding site. This is contrary to covariations in protein-protein interactions, where the epistasis comes from the constrains from direct physical interactions (packing, h-bonds, etc). If we neglect the direct interactions between bases then, the epistasis found may suggest that more than one factor would bind there to enhance the final expression. This may be more intense in stronger promoters. A discussion about this point may benefit the discussion.
- Some Supplementary Information Tables of large size would be more useful in Excel format.

Reviewer #2 (Remarks to the Author):

Yus et al. present a novel approach to rapidly identify optimal gene expression initiation signals using the highly dynamic response of DNA adenine methylase as reporter that can be read out in a high throughput fashion. They apply this method to the identification of optimal signals for *Mycoplasma pneumoniae*.

Major point:

I feel the actual work is brilliant, but the presentation leaves room for improvement. Actually, it was not clear to me what was the real aim of this study unless I had read the first part of the results. The rationale and the direct aim of the procedure need to be explained much better with the non-specialized reader of Nature Comms. in mind. Accordingly, the Summary and Introduction should be newly written.

Minor points:

The abbreviation TSS does not seem to be defined.

l. 7: delete bracket after "sequencing"

l. 15: No need to mention researchers and companies in the abstract

l. 29: "*Synechococcus*" should appear in italics

l. 46: The sentence starting with "In fact, being ..." is not clear to me. Please re-phrase.

l. 50-58: This § is not sufficient to understand the principle of the method.

I. 134: The authors might wish to refer to the first systematic promoter study in *M. pneumoniae* that has already shown the weak relevance of the -35 region (Halbedel et al., 2007).

Reviewer #3 (Remarks to the Author):

The ability to predict the levels of transcription from a specific promoter or the efficiency of translation from a specific translation start site are important to enable the design of new bacterial circuits in synthetic biology efforts. Likewise, accurate means to measure these parameters are important for basic studies into molecule processes in cells. In bacteria, these predictions and measurements are complicated by the fact that different bacteria have different sequences determinants, and in vivo measurements are often lacking in sensitivity, dynamic range, and the ability to assess these rates globally.

In this manuscript, Yus et al present a method in which they coupled the ability of the Dam enzyme to methylate DNA in a dose-dependent manner with high-throughput sequencing to measure the transcriptional and translation efficiencies of a comprehensive set of sequences in *M. pneumoniae*. The method depends on the idea that the amount of Dam in a cell correlates with the amount of DNA methylation at installed GATC sites. The Dam thus serves as a reporter of the transcription or translation efficiency of a given sequence.

Overall, this is a very clever idea, and the authors have clearly demonstrated its usefulness in this bacterial species. The results correlate with known behavior, but also expand on known mechanisms and, most importantly, have predictive power. This method can now be easily extended to many more bacterial species to build a comprehensive database of transcriptional and translation efficiencies in diverse bacteria. In turn, this enables more precise design of circuits for synthetic biology and for optimizing expression of RNA and proteins in bacteria, in general.

I have little to criticize here, and overall support publication.

REVIEWERS' COMMENTS:

Reviewer #1 (Remarks to the Author):

Outstanding manuscript by Yus et al. on a new experimental and computational technique to quantify gene expression in vivo. It could be applied to many organisms and the authors validated it on an idiosyncratic organism such as *M. genitalium*, where the signal sequences for the initiation of transcription and translation are non-standard. The method relies on the in vivo expression of a DNA methylation enzyme followed by deep sequencing and data analysis using probabilistic and information theory models. The forward engineering of regulatory sequences is one of the limiting step in synthetic biology, where there will be a shift from characterizing existing biological parts (such as promoters and RBSs) to their de novo design. Many labs currently use more often synthetic RBSs libraries (obtained through computational design) than available known RBS. The reason for this paradigm shift is the lack of modularity of such parts and the dependency on the context. The Yus et al. manuscript uses a simple experimental technique to produce predictive models for both transcription and translation initiation. The data, Figures, Tables and text are of excellent quality. I recommend to publish the manuscript in Nature Communications after replying to the minor questions suggested below.

Questions:

1. *Intermediate levels of dam expression will create hemimethylations that will be counted as methylated, therefore overestimating the dam expression. Is this true? Of course, this systematic bias could be accounted in modelling. The discussion may benefit from a clarification on this.*

Response: We would like to thank the Reviewer for his or her comments. As far as we know, neither DpnI nor MboI can effectively cleave hemimethylated GATC sites. DpnI cuts hemimethylated DNA 60-100 times less efficiently than fully methylated GATC sites (See Supplementary Note 2). MboI is also known to cut hemimethylated DNA with less efficiency (3%) or to not cut hemimethylated DNA at all (See Supplementary Note 2). As such, hemimethylated GATC does not interfere with our results since DpnI and MboI un-cut read counts would be nearly the same.

Indeed, intermediate levels of Dam expression will create not only hemimethylated DNA but also partially methylated stats of the four GATC sites (only some of the four GATC sites will be fully methylated). In this case, both DpnI and MboI can cut partially methylated GATC sites, and consequently we might not be able to retrieve sequences generating intermediate levels of Dam expression. We have simulated these hemimethylated and partially methylated DNA effects, and described the results in Supplementary Note 2. In summary, there is a greater chance to obtain partially methylated DNA as the number of GATC sites increases. This enables us to enrich for 'extreme' sequences that either highly or lowly produce Dam. This is because the sequences that lead to intermediate levels of Dam expression can be cut by both DpnI and MboI,

and consequently cannot be amplified in the following PCR. On the other hand, we found that hemimethylated DNA can increase the number of these reads, allowing us to obtain sequences producing intermediate levels of expression.

We have added two figures (4 and 5) to Supplementary Note 2 with the simulations. We have also added the following paragraph to the main text:

p.6-7 paragraph: “In the event of hemimethylation, GATCs cannot be cut by either DpnI or MboI. We simulated the effect of hemimethylation (Supplementary Note 2) and showed that it does not significantly affect the results.”.

2. In principle, here any sequence region that would be a binding region for more than one protein would produce epistasis between all the nucleotides within the binding site. This is contrary to covariations in protein-protein interactions, where the epistasis comes from the constraints from direct physical interactions (packing, h-bonds, etc). If we neglect the direct interactions between bases then, the epistasis found may suggest that more than one factor would bind there to enhance the final expression. This may be more intense in stronger promoters. A discussion about this point may benefit the discussion.

Response: We would like to thank the Reviewer for the insightful comments. We agree with the Reviewer’s point of view about epistasis. There could be three types of epistasis. One as the reviewer suggests, in which two or more proteins could bind to different regions of the promoter and have a cooperative interaction favoring or disfavoring transcription or translation. The second in which a protein with several domains (i.e., sigma 70) could bind in a cooperative manner to different regions of the promoter, and therefore non-adjacent sequences would have an epistatic effect. The third one in which a single domain binding to several bases would have a network of cooperative interactions (e.g., H-bonds, hydrophobic interactions, conformational entropy...), causing adjacent bases to have an epistatic interaction. In the case of *Mycoplasma pneumoniae* and looking at the distance between epistatic interactions, it is more likely that we have the second and/or third scenarios. We have commented on this on the MS.

3. Some Supplementary Information Tables of large size would be more useful in Excel format.

Response: We would like to thank the reviewer for the useful suggestions. We have now uploaded some Excel files as supplementary data.

Reviewer #2 (Remarks to the Author):

Yus et al. present a novel approach to rapidly identify optimal gene expression initiation signals using the highly dynamic response of DNA adenine methylase as reporter that can be read out in a high throughput fashion. They apply this method to the identification of optimal signals for *Mycoplasma pneumoniae*.

Major point:

1. *I feel the actual work is brilliant, but the presentation leaves room for improvement. Actually, it was not clear to me what was the real aim of this study unless I had read the first part of the results. The rationale and the direct aim of the procedure need to be explained much better with the non-specialized reader of Nature Comms. in mind. Accordingly, the Summary and Introduction should be newly written.*

Response: We would like to thank the reviewer for the useful suggestions. We have now more clearly stated the aim of this study in both the abstract and the introduction.

Minor points:

The abbreviation TSS does not seem to be defined.

l. 7: *delete bracket after "sequencing"*

l. 15: *No need to mention researchers and companies in the abstract*

l. 29: *"Synechococcus" should appear in italics*

l. 46: *The sentence starting with "In fact, being ..." is not clear to me. Please re-phrase.*

l. 50-58: *This § is not sufficient to understand the principle of the method.*

l. 134: *The authors might wish to refer to the first systematic promoter study in *M. pneumoniae* that has already shown the weak relevance of the -35 region (Halbedel et al., 2007).*

Response: We have changed the main text according to reviewer's suggestions.

Reviewer #3 (Remarks to the Author):

The ability to predict the levels of transcription from a specific promoter or the efficiency of translation from a specific translation start site are important to enable the design of new bacterial circuits in synthetic biology efforts. Likewise, accurate means to measure these parameters are important for basic studies into molecule processes in cells. In bacteria, these predictions and measurements are complicated by the fact that different bacteria have different sequences determinants, and in vivo measurements are often lacking in sensitivity, dynamic range, and the ability to assess these rates globally.

In this manuscript, Yus et al present a method in which they coupled the ability of the Dam enzyme to methylate DNA in a dose-dependent manner with high-throughput sequencing to measure the transcriptional and translation efficiencies of a comprehensive set of sequences in *M. pneumoniae*. The method depends on

the idea that the amount of Dam in a cell correlates with the amount of DNA methylation at installed GATC sites. The Dam thus serves as a reporter of the transcription or translation efficiency of a given sequence.

Overall, this is a very clever idea, and the authors have clearly demonstrated its usefulness in this bacterial species. The results correlate with known behavior, but also expand on known mechanisms and, most importantly, have predictive power. This method can now be easily extended to many more bacterial species to build a comprehensive database of transcriptional and translation efficiencies in diverse bacteria. In turn, this enables more precise design of circuits for synthetic biology and for optimizing expression of RNA and proteins in bacteria, in general.

I have little to criticize here, and overall support publication.

Response: We would like to thank the Reviewer for his or her positive assessment of this work.